# Observing Liquid Sloshing Based on a Multi-Degree-of-Freedom Pendulum Model and Free Surface Fluctuation Sensor

**DOI:** 10.3390/s23218831

**Published:** 2023-10-30

**Authors:** Xiaojing Qi, Yingchao Zhang, Bolin Gao

**Affiliations:** 1School of Vehicle and Mobility, Tsinghua University, Beijing 100084, China; komasaqi@foxmail.com; 2College of Automotive Engineering, Jilin University, Changchun 130022, China; yingchao@jlu.edu.cn

**Keywords:** liquid sloshing, observer, multi-DOF pendulum model, free surface fluctuation sensor

## Abstract

Rollover prevention of partially filled tank trucks is an ongoing challenge in the road transportation industry, with the core challenge being real-time perception and observation of the liquid state inside the tank. In order to realize reliable observation of a sloshing liquid, this article first proposes a sloshing modeling method based on a multi-degree-of-freedom pendulum model and derives the double mass trammel pendulum model (DMTP, 2DOF) accordingly, which accurately reflects the sloshing dynamics under wider operating conditions. Second, a free surface fluctuation sensor is designed based on magnetostriction, capable of measuring the inclination and height of the liquid level inside tanks filled with hazardous chemicals. Finally, the unscented Kalman filter (UKF) is utilized to synthesize the information of the two, establishing a credible real-time observation of the sloshing liquid. Verified using a vehicle–fluid coupled co-simulation, under the condition of a consecutive double lane change, the observation error of the proposed method is only 25.9% of that of the open-loop calculation, providing a secure guarantee for the observation of the state variables of the single pendulum model (SP) used for most kinds of anti-rollover control.

## 1. Introduction

Tank trucks account for about 18% of the total number of commercial trucks and play a significant role in highway transportation; they also account for over 30% of truck rollover accidents [1], attributed to their poor rollover stability and caused not only by their high center of mass or heavy loading but also the coupling of the roll movement of the vehicle and a sloshing liquid [2,3]. Moreover, with 80% of hazardous liquid chemicals, such as gasoline, diesel, chlorinated hydrocarbons, strong acids, etc., relying on tank trucks to be transported on highways [4], severe accidents ensue once rollover occurs. Therefore, anti-rollover control of tank trucks is essential, for which the main technical approaches are establishing a more accurate surrogate model based on a sloshing mechanism and designing anti-rollover control algorithms considering the sloshing effect, as well as measuring or estimating the required state variables for control by designing sensors and observers.

Existing models describing sloshing liquid in tanks can be generalized into three categories: quasi-static (QS) models, mechanical equivalent models, and fluid dynamics models [5].

QS models ignore sloshing dynamics and only model the static-state position of the liquid’s center of mass based on the tank’s geometric cross-section [6,7,8,9], thus having low accuracy in transient conditions.

Mechanical equivalent models approximate the force output characteristics of a liquid-filled system by establishing a mechanical model that meets the equivalence principle [10] with the original system. The commonly used models include single pendulum (SP) [11,12,13], spring-mass-damper [14], particle–cluster–rod [15], and trammel pendulum (TP) [16,17,18,19]. Most mechanical equivalent models perform well in terms of accuracy and solving speed under dynamic conditions and can reflect first-order sloshing, and thus can be used for real-time control. However, they usually neglect higher-order sloshing such as hydraulic jump and splash, so they can only describe approximately linear sloshing well.

Fluid dynamic models, including models based on potential flow theories and finite element models based on Navier–Stokes equations (referred to as the CFD method in the following text), etc., can be used, but potential flow theory [20,21] can only be applied for certain tanks with specific section geometries under linear sloshing. Although the finite-element-based CFD method [22,23,24,25,26] is almost accurate under various transient conditions, it requires a daunting amount of computation, and thus cannot be used for real-time control.

Although not suitable for usage in real-time control, the CFD method is suitable for parameter calibration of mechanical models and offline co-simulation for control algorithm verification, with its results reckoned as true values. Efforts have been made in the CFD field to better simulate sloshing behavior. Tang et al. [27] found that compared with the LES method, using the k-ε model reduces the computing time by 40% under similar error rates for liquid sloshing in 3D LNG tanks. Ganuga et al. [28] established a three-dimensional, fully coupled FSI model, which revealed that the behavior of internal structures in a sloshing tank is subjected to resonance, highlighting the role of flexible baffles. Jadon et al. [29] used an integrated experimental and multi-physics numerical approach to predict automotive fuel tank sloshing noise by simulating the fluid sloshing, dynamic forces, vibration displacement, and noise radiation. Zheng et al. [30] proposed the ISPH method to accurately simulate violent sloshing with complex baffles, providing insights into effective engineering solutions for dampening sloshing and reducing impact pressures.

For controller and observer design, real-time computation is essential, so the most commonly used models are various pendulum models in mechanical equivalent models, especially the SP model, whose accuracy directly affects the performance of the controller and observer.

However, the pendulum described in the model has no real-world equivalent other than liquid, so the swing state of the pendulum cannot be directly measured. Therefore, it is necessary to establish an observer to estimate and update the state variables of the pendulum model for the controller according to the measuring of the dynamics of the free surface of the liquid. The collection of dynamic liquid-level data is usually based on cameras [31,32,33] or ultrasonic sensors [34]. However, it is not proper to apply transparent tanks in reality, precluding measurement based on visual information. Furthermore, ultrasonic sensors need to be as perpendicular as possible to the measuring interface, making it almost impossible to utilize them for a sloshing liquid.

In this article, a multi-degree-of-freedom (multi-DOF) pendulum model is proposed to address the limited range of applicable working conditions of existing mechanical equivalent models. Furthermore, a free surface fluctuation sensor is designed based on magnetostriction, and an unscented Kalman filter (UKF) is applied as an observer to synthesize the multi-DOF pendulum model and sensor data, which solves the problem of model phase mismatch and provides an accurate state estimation for the equivalent pendulum model (generally the SP model) used for the anti-rollover control of tank trucks.

## 2. Spectral Analysis of Sloshing Liquid

### 2.1. CFD Modeling of Liquid Sloshing

To accurately model the liquid sloshing phenomenon in the tank, we used a precise finite volume-based computational fluid dynamics (CFD) method to simulate the gas–liquid two-phase flow inside the tank. Figure 1 illustrates our research focus, which is the lateral sloshing behavior of an elliptical section tank found in a semi-trailer tank truck. The tank has dimensions of 1240 mm for the long axis and 900 mm for the short axis.

Considering the legal requirements that mandate the installation of longitudinal wave-proof plates at specific intervals, as well as the fact that the longitudinal acceleration is relatively negligible compared to the lateral acceleration during cruising, we solely focused on the sloshing within the trailer’s roll plane. Consequently, for the sake of computational efficiency and by neglecting longitudinal sloshing effects, we used a 2-dimensional modeling approach for the two-phase flow using the CFD software Star-CCM+ 2021.3 (16.06.008-R8).

In line with the typical sloshing velocity near the wall and using an empirical boundary layer selection method, we established a 3 mm thick boundary layer consisting of five sub-layers of grids. Following a grid independence analysis, we ensured that the maximum side length of the grid globally was controlled within 10 mm while maintaining a minimum length of approximately 5 mm. This approach guaranteed the accurate capture of the dynamic state of liquid sloshing within the tank’s scale.

The primary focus of this study is on low-viscosity light liquids that resemble gasoline or diesel. In such fluids, the flow demonstrates significant turbulent behavior due to high Froude and Reynolds numbers. In these scenarios, volume forces dominate the flow rather than viscosity. Consequently, the influence of viscosity is relatively small, rendering the system robust against variations in liquid viscosity parameters. Liquids like water, gasoline, and diesel exhibit similar oscillation characteristics when subjected to sloshing in tanks, with comparable frequencies across different modes of oscillation.

In our research, we used gasoline and air as the medium for the simulation. The parameters for the two Euler phases are detailed in Table 1, while other relevant continuum parameters are provided in Table 2.

In terms of solver settings, an implicit unsteady solver with adaptive time stepping is used to balance the computational efficiency and solution time for a large number of simulations. The volume of fluid (VOF) method is used to track the interface fluctuations between the gas and liquid phases. The k−ϵ turbulence model is utilized for solving, according to [27]. The implicit unsteady solver has a time step of 0.005 s, and the minimum step size for the adaptive time step is set as $10^{-5}$ s to ensure good convergence while maintaining fast simulation speed.

### 2.2. Spectral Analysis of Liquid Sloshing

According to the concept of spectral analysis, the sloshing of liquid can be decomposed into an approximate linear combination of its 0- to ∞-order sloshing mode. An illustration of sloshing modes of liquid is shown in Figure 2a, where the zero-order sloshing embodies the inertia of fluid as a rigid body, the first mode reflects the sloshing planar free surface, and the second mode and higher-order sloshing depict the sinusoidal fluctuation of free surface. The higher the order of sloshing, the less liquid participates but characterizes more details.

To better understand the characteristics of the output force of sloshing liquid in the tank, CFD simulations are carried out under three conditions: lateral acceleration step excitation $a_{y}=1,3,5\;m/s^{2}$, filling rate f=50%, and the fast Fourier transform (FFT) of side-force output. Lateral force output is shown in Figure 2b, where the amplitude and nonlinearity of sloshing increase with $a_{y}$, and hydraulic jump and splash also become obvious. From Figure 2c, it can be seen that the lateral force output shows more high-frequency components with an increase in $a_{y}$. According to Figure 2d, it is found that in addition to the first mode, the second and higher-order modes cannot be ignored, which explains why it is difficult to thoroughly describe liquid sloshing using linear models, as they ignore the higher-order sloshing components.

## 3. Multi-Degree-of-Freedom Pendulum Model

### 3.1. Generalized Pendulum Model

The most commonly used models to describe liquid sloshing are various pendulum models. Based on these models, the generalized pendulum model is proposed, as illustrated in Figure 3. This model comprises single pendulums, trammel pendulums, and all possible linear and/or nonlinear combinations of them. Yang et al. introduced a 1DOF multi-mass trammel pendulum to simulate liquid sloshing in an elliptical sectioned tank [35,36]. However, this model did not increase the degrees of freedom (DOF) of the pendulum; it only split and redistributed the concentrated swing mass, as depicted in Figure 3c. Consequently, it exhibits no substantial difference in terms of swing mode compared to a trammel pendulum with one swing mass. Different pendulum models vary in the number of swinging DOF and the number of concentrated masses, which allows for diverse potential in describing liquid sloshing. Theoretically, a higher number of DOF enhances the fitting ability of the model but also increases the risk of overfitting and requires more computational resources. Therefore, it is crucial to select an appropriate generalized pendulum model that balances computational complexity and accuracy.

### 3.2. Spectral Analysis of Models

A spectral analysis was conducted on various sloshing models, with results shown in Table 3. The rigid body model can only embody the inertia of the rigid body (zero-order sloshing), whereas the QS model can express the inertia together with the free surface inclination angle (and the movement of the liquid center of gravity caused by it), but it cannot express the sloshing dynamic. A linearized SP model can only express up to the first mode; SP and TP can express the second and third modes, but their accuracy is low for higher-order sloshing. A multi-DOF pendulum model can express high-order sloshing, but its accuracy is not as accurate as that of the finite volume method-based CFD model (FVM-CFD), and the latter is too clumsy to compute in real-time. Therefore, in order to obtain a more accurate model while balancing computational complexity, the DOF of the pendulum model can be appropriately increased.

### 3.3. Pendulum Models with 2DOF

On the basis of the existing SP and TP models, dual-mass trammel pendulum (DMTP) and combined trammel and single pendulum (TPSP) models with 2DOF are proposed, as shown in Figure 4, which are nonlinear and linear combinations (series and parallel) of SP and TP, respectively. According to the Lagrange equation, their dynamic model can be deduced, and the details can be found in Appendix A. Utilizing the derived function, simulations are carried out in the following sections.

## 4. Simulation Analysis of Pendulum Models

### 4.1. Parameter Fitting for Pendulum Models

To obtain the parameters of the pendulum model and compare SP and TP models, as well as the proposed DMTP and TPSP models, a lateral acceleration step excitation CFD simulation dataset: dataset No.1, as shown in Figure 5, was established, which includes 30 sets of simulation data, reflecting the free sloshing of liquid. Secondly, the loss function representing model fitness is defined in (1).
(1)$$J={\left\|\frac{F_{y,a_{y}}-F_{y0,a_{y}}}{\bar{F_{y0,a_{y}}}}\right\|}_{2}^{2}+{\left\|\frac{F_{z,a_{y}}-F_{z0,a_{y}}}{\bar{F_{z0,a_{y}}}}\right\|}_{2}^{2}{+\left\|\frac{M_{x,a_{y}}-M_{x0,a_{y}}}{\bar{M_{x0,a_{y}}}}\right\|}_{2}^{2},\;a_{y}=1,2,3,4,5$$
where $\bar{F_{y0,a_{y}}}$, $\bar{F_{z0,a_{y}}}$, and $\bar{M_{x0,a_{y}}}$ are the time average values of $F_{y}$, $F_{z}$, and $M_{x}$. $F_{y,a_{y}}$, $F_{z,a_{y}}$, and $M_{x,a_{y}}$ are the lateral force, vertical force, and roll moment output of the model under step excitation of $a_{y}$.

The genetic algorithm is applied to optimize the loss function so as to fit the optimal structural parameters of the model. To ensure convergence, each model is recalculated eight times at each filling rate, and the optimal result is taken as the fitting parameter. The optimization results are shown in Figure 6. SP is better than TP at medium to high filling rates, whereas DMTP is the optimal model under all examined filling rates, followed by TPSP, reflecting the higher fitting potential of the two proposed 2DOF models compared with conventional 1DOF models.

### 4.2. Map Analysis of Pendulum Models

To verify the generalization ability of models in other working conditions, firstly, another CFD simulation dataset: dataset No.2 with 150 sets of data is established, as shown in Figure 7, with lateral acceleration sinusoidal excitation to reflect the forced sloshing of liquid.

On top of that, the orthogonal conditions of sinusoidal excitation $a_{y}$ with different amplitude and frequency are verified on datasets No.1 and No.2, and the normalized errors of force outputs $F_{y,a_{y}\;}$, $F_{z,a_{y}\;}$, and $M_{x,a_{y}\;}$ of the four pendulum models calculated using the optimal fitting parameters are compared with the CFD results. The errors are shown in Figure 8. The larger and darker the blue area, the better the model conforms to more working conditions. DMTP is the optimal model at medium to low filling rates (f≤60%), whereas at higher filling rates, the sloshing effect attenuates, and all models achieve almost the same performance; therefore, the optimal model in this case is SP, which requires the least computation. To some extent, the advantages of DMTP as a more accurate pendulum model have been verified.

DMTP is a nonlinear pendulum model with more computation than traditional 1DOF models, but it can still be real-time computed even on an STM platform for general use with a main frequency of only 24 MHz. Although limited by the high operation frequency of model predictive controllers, it would be difficult to use DMTP as a predictive model in MPC controllers, but it is proper to use it as the system motion equations in the observer with a lower frequency (about 1000 times of simulation per second and below) to improve observation accuracy.

## 5. Sloshing Observer Design

### 5.1. Free Surface Fluctuation Sensor Design

As mentioned in Section 1, considering computational burden, existing research typically uses linearized SP within the controller to describe liquid sloshing so as to exert anti-rollover control. However, linear models are not precise enough and thus require external measurements or estimates to unremittingly update model states. As shown in Figure 9a, state variables of the vehicle model are all available using direct/indirect measurements of existing sensors or estimated, like lateral velocity $v_{y}$ (or the sideslip angle β, $tan\beta =v_{y}/v_{x}$) [37,38,39]. Nevertheless, state variables related to liquid (pendulum) models, such as the swing angle θ and angular velocity $\dot{\theta }$, can neither be obtained with existing sensors nor estimation from available vehicle states since information on a liquid in vehicle states is limited; therefore, theoretically, an unrealistic extremely accurate vehicle model is needed to estimate liquid state.

Existing works invoking pendulum model assumptions to control vehicles do not involve real vehicle tests, with only simulations in which pendulum substitute liquid were carried out [12,13,18,19]. In addition, state variables (e.g., the swing angle θ) are directly fed back to the controller for state updates, ignoring the fact that instead of a pendulum, there exists only liquid in the actual tank, which is currently unable to be measured.

The error characteristics of the open-loop calculation of the models are shown in Figure 9b. After a single maneuver, such as steering or a lane change, starting from a stable state, the response can be divided into 0–3 stages based on the error characteristics between the open-loop model and the actual liquid trilateral force output.

Stage 0: In this stage, when the vehicle is in a stable state, there is minimal fluid sloshing, and the lateral acceleration is almost zero. The difference between the model and the actual system is negligible.

Stage 1: When a maneuver is initiated from a stable state, the fluid undergoes forced sloshing. As the natural sloshing frequency of the model and the real system may not match exactly, phase errors start to accumulate gradually. However, during forced oscillation, the system output closely follows the variations in the input frequency, with minimal impact from the natural frequency. Therefore, the output error increases over time, but at a relatively slow rate.

Stage 2: Once the maneuver is completed, the vehicle returns to stable straight-line motion, and the fluid enters free sloshing. In this stage, with no external forces acting on the system, its oscillation is solely determined by its natural frequency. Consequently, the output error rapidly increases with time.

Stage 3: Due to the presence of fluid damping, both the model and the actual system tend to stabilize under the damping effect. The error gradually decreases, and eventually, the system returns to Stage 0, where the discrepancies between the model and the actual system are minimal once again.

Throughout this process, the errors are not significant enough to cause controller failure. However, if another maneuver is initiated without allowing the system to settle after completing a previous operation, while the system is in Stage 2 or Stage 3, the errors quickly accumulate, potentially leading to model mismatch, which further leads to controller failure and eventually rollover. Therefore, a free surface fluctuation sensor is designed to measure the liquid state so as to estimate the states of a pendulum model.

As shown in Figure 10, a magnetostrictive liquid level meter constitutes the main part of the fluctuation sensor. Floating balls containing a permanent magnet slide on the measuring rod, and the level meter functions by timing the round-trip of torsional stress wave from the digital head to the magnets in balls.

The existing level meter is capable of simultaneously measuring the positions of multiple floating balls and reaches an accuracy of 0.5%/0.3 mm (larger one), and the maximum measurement frequency reaches 2500–4000 Hz.

Based on the level meter, a rocker arm as well as a retractable floating board are complemented to form a measurement triangle, as shown in Figure 10b. Parameters of the measuring triangle are selected in advance, and the distance between the upper and lower floating balls differs with the liquid-free surface inclination angle. Using the law of cosines in (2)
(2)$$\theta_{surf}=\cos^{-1}\left(\frac{r^{2}+x^{2}-l^{2}}{2rx}\right)-\frac{\pi }{2}$$
the inclination angle fitted by the floating board can be obtained, where $\theta_{surf}$ is the average inclination of the liquid surface measured with the sensor, r is the center distance between the upper floating ball and the hinge on one side of the central float board, x is the distance between the upper and lower floating balls measured with the sensor, and l is the designed length of the rocker’s arm.

The common range of surface inclination is generalized by analyzing simulation results. It is found that under large acceleration step excitation ($a_{y}=5\;m/s^{2}\approx 0.51\;g$), the liquid level inclination does not exceed ±15°, and in reality, such a large lateral acceleration is scarce. For safety consideration, ±20° is chosen as the measurement range. Due to regulatory limitations [2,3], the maximum filling rate must not exceed 95%, and since the impact of liquid sloshing at high filling rates can even be ignored, in order to ensure a measurement range of ±20° at the highest design filling rate f=90% and cover at least one-half of the free surface width at the lowest design filling rate f=50% to filter out the impact of the third mode and higher-order sloshing, the geometric parameters of the floating board are designed, as shown in Figure 10c.

As in Figure 10d, by using a lightweight pulley block with a specific magnification, the floating board length can be automatically adjusted according to the filling rate under the balance of the buoyancy of the floating board and the elastic restoring force of the recovery spring.

Under general sloshing conditions that do not cause severe breakage of the liquid surface, the swing angle θ of SP in the controller can be approximated as the linear transform of $\theta_{surf}$, i.e.,
(3)$$\theta =K\theta_{surf}$$
where K is a constant.

### 5.2. Observer Based on DMTP and Senser Data

According to the accurate nonlinear DMTP model and the data of the fluctuation sensor, an observer based on an unscented Kalman filter (UKF) can be designed to estimate the swing angle of the linearized SP required in the controller, as shown in Figure 11.

#### 5.2.1. Fusion of Angles of DMTP to Linearized SP

From Section 4, it can be seen that the open-loop accuracy of DMTP is optimal at medium filling rates and below (f≤60%), but linearized SP in the controller has only one DOF, i.e., θ. Similarly, the fluctuation sensor can only measure the average inclination angle $\theta_{surf}$, so identifying how to equal the angles of two DOF of DMTP, i.e., $\theta_{1},\theta_{2}$, to θ is an urgent problem.

There are two methods for fusing two angles, as shown in Figure 12b,c.

The center of mass conversion method takes the angle of C.G. of $m_{1}$ and $m_{2}$ in DMTP as θ, i.e., (4) and (5),
(4)$$k_{m}=\frac{m_{2}}{m_{1}+m_{2}}$$
(5)$$\theta_{eql}=\tan^{-1}\frac{lsin\theta_{1}+k_{m}r\sin\left(\theta_{1}+\theta_{2}\right)}{lcos\theta_{1}+k_{m}r\cos\left(\theta_{1}+\theta_{2}\right)}$$

The convex combination method assumes θ as the convex combination of $\theta_{1}$ and $\theta_{2}$, i.e., (6),
(6)$$\begin{array}{c} \theta_{eql}=k_{conv}\theta_{1}+\left(1-k_{conv}\right)\theta_{2} \end{array}$$
where the combination proportion $k_{conv}\in \left[0,\;1\right]$ is determined by the relationship between $m_{1}$ and $m_{2}$, as in (7),
(7)$$k_{conv}=k_{m}=\frac{m_{2}}{m_{1}+m_{2}}$$
or in (8),
(8)$$k_{conv}=\frac{m_{1}+m_{2}}{m_{1}+2m_{2}}$$

By simulating the two methods above in extreme maneuvering conditions, i.e., left cornering, (high speed) single lane change, and (short distance) double lane change, on the vehicle–fluid coupling co-simulation platform, which will be constructed in Section VI, A, the equivalent fusion angle $\theta_{eql}$ is obtained. The value is then input into force output calculating equations for SP. The results are shown in Figure 12d,f.

It can be concluded that the convex combination method with (6) and (8) is the most apropos one, since the time average error of force outputs ($F_{y}$, $F_{z}$, and $M_{x}$) of the whole simulation process are the lowest (corresponding to the final value of the last inset in Figure 12d,f), and the fusion results are superior to that of the open-loop calculation of SP.

#### 5.2.2. Angle Decomposition and UKF Construction

Before harnessing DMTP, there are still two essential questions to address.

To start with, the question is how can the angle $\theta_{surf}$ measured with the fluctuation sensor be decomposed into two angles $\theta_{1}$ and $\theta_{2}$ in DMTP in the process of inputting sensor data in order to correct the model state with measurement. The solution is simple but intuitive: make the best guess of $\theta_{1}$ and $\theta_{2}$ based on measurement $\theta_{surf}$. The exact process is: now that we have the values of $\theta_{1}$ and $\theta_{2}$ for the last discrete time step, these values can be denoted as known values $\theta_{1,k-1}$ and $\theta_{2,k-1}$. Furthermore, the relationship between the current values required to estimate $\theta_{1,k}$, $\theta_{2,k}$, and the known measurement $\theta_{surf}$ was already derived in (6) and (8). Then, a convex optimization problem for $\theta_{1,k}$ and $\theta_{2,k}$ can be constructed to find $\theta_{1,k}$ and $\theta_{2,k}$ with the least deviation to $\theta_{1,k}$ and $\theta_{2,k}$ while subject to the relationship described in (6) and (8). The details of the solution derivation can be found in Section B.1.

The second question is how can the model nonlinearity be addressed when putting it into a filtering algorithm. The answer is to use an unscented Kalman filter since the equations for system motion involve DMTP, which is difficult to linearize, and UKF does not even need the analytical form of the motion and observation equations as it simply regards the system as a black box. In the solution of this paper, UKF is applied for observer construction. The construction of UKF is contained in Section B.2.

UKF cannot guarantee convergence in general nonlinear systems, but its iterative version, the iterated unscented Kalman filter (IUKF), can theoretically converge to the posterior mean of the true value. However, due to the assumption that the observation equation is linear as (3), IUKF is equivalent to UKF. In theory, UKF converges to the posterior mean of the equivalent pendulum angle of linearized SP.

In practical applications, it is found that there exist some local fluctuations in the sensor data that are difficult to filter out, which will cause an impact when calculating the force output. Therefore, a Luenberger observer is appended after the sensor data to smooth it, and its two poles are designed as [–5, –5].

## 6. Simulation and Analysis of Sloshing Observation

### 6.1. Simulation Model

To better validate the effectiveness of the observer constructed in the last section, a vehicle–fluid coupling co-simulation verification platform is built, as shown in Figure 13.

By writing a MATLAB function and Java macro, Simulink and StarCCM+ are controlled to read and write data automatically in the simulation process, and two flag-bit sharing files and two data exchange sharing files are constructed to realize the co-simulation of vehicle dynamics and finite element-based CFD. With the help of this simulation platform, real-time interaction between vehicles and fluid dynamics can be more realistically simulated.

A simulation model is built in Simulink, as shown in Figure 14, to validate the proposed observation method based on a multi-DOF pendulum model (specifically, DMTP model) and the free surface fluctuation sensor.

### 6.2. Acquisition of Fluctuation Sensor Data

Currently, no physical prototype of a fluctuation sensor is available; therefore, simulation methods are used to simulate it. As shown in Figure 15, sensor data are obtained with image processing using CFD simulation images output by Star-CCM+. The specific method is:

Step 1: At each simulation moment, Star-CCM+ outputs an image of liquid shaking.

Step 2: Grayscale the image and extract edges.

Step 3: Mask the parts of the image irrelevant to the free liquid level and trim the free surface according to the length of the floating board.

Step 4: Obtain equivalent sensor data $\theta_{surf}$ by fitting scatter points of the trimmed liquid surface with a straight line.

### 6.3. Results Analysis

As analyzed in Figure 9b, the error of the open-loop calculation of the model accumulates swiftly under continuous maneuvering conditions, making it the most effective condition to test the performance of the proposed observation method. Therefore, the consecutive double-lane change condition is designed, as shown in Figure 16a. The reference path is two double-lane change paths with an interval of only 100 m. Each double-lane change path completes a lane change within 45 m, with a 45 m straight line, and a change back within another 45 m. The initial speed of the semi-trailer tank truck is 70 km/h.

The simulation results are shown in Figure 16b,g, which were obtained using the lateral acceleration a of the trailer $a_{y2}$ and roll angular velocity ${\dot{\varphi }}_{2}$ from TruckSim (in real vehicles, they can be measured using the IMU fixed at the bottom of the liquid tank). White noise was added to $a_{y2}$ and ${\dot{\varphi }}_{2}$ with a variance of 0.001 to reflect the measured noise of IMU as the input of the UKF equations of system motion, and the fluctuation sensor measured result $\theta_{surf}$ is supplemented for state estimation.

Figure 16e compares the angle fusion results with the angle calculated using open-loop SP and DMTP. Figure 16f compares the roll moment $M_{x}$ obtained by inputting the fusion results and the equivalent angle calculated/estimated using open-loop SP into the force calculation formula of SP and the roll moment output of CFD as a true value for observer validation.

In Figure 16g, the time average error of the predicted roll moment is compared between the proposed observation method (data fusion method) and the open-loop method with CFD results. The error at each moment is the average of all previous moments’ errors, and the value at the last moment represents the mean error throughout the entire simulation process.

Before the 10th second, the open-loop model exhibits sufficient accuracy. However, as the first double lane change concludes, the liquid in the tank enters a free-sloshing state. At this point, the slight difference between the model’s natural frequency and that of the actual liquid-filled system begins to manifest, leading to an accumulation of phase differences. Before the damping effect of the system can eliminate the accumulated errors, the vehicle initiates the second double-lane change, causing a rapid accumulation of errors in the open-loop calculation.

However, the proposed observation method, which incorporates fluctuation sensor data, consistently maintains high measurement accuracy. The time mean observation error during the entire process is only 25.9% of the time average error of the open-loop calculation, significantly improving the observation accuracy by about four times. Particularly, during the second maneuver, the instantaneous error is only about one-seventh of that observed in the open-loop calculation.

## 7. Conclusions

In order to realize reliable observation of states of the pendulum model (reflecting liquid sloshing dynamics) for anti-rollover control of tank trucks, to begin with, this article proposes a modeling method for liquid sloshing of tank trucks based on a multi-degree-of-freedom (multi-DOF) pendulum model. We conducted a spectral analysis of the output force characteristics of liquid sloshing in the tank and designed a double mass trammel pendulum model (DMTP, 2DOF) and a combined trammel and single-pendulum model (TPSP, 2DOF) that reflect the sloshing dynamics more accurately, especially under medium and low filling rates (f≤60%). Using simulations under orthogonal working conditions of step and sinusoidal lateral acceleration excitation, DMTP and TPSP are proven to have advantages over conventional one-degree-of-freedom pendulum models (i.e., SP and TP) in terms of generalization ability and fitting precision.

In addition, a free surface fluctuation sensor is designed that can measure the inclination and average height of the sloshing liquid surface. It includes a retractable floating board that can automatically adjust its length according to changes in the filling rate. The characteristic that the measurement part has no electronic components makes it suitable for applications in tanks containing hazardous fluid chemicals. Furthermore, an unscented Kalman filter (UKF) is applied to observe the liquid sloshing.

In the end, using the constructed vehicle-fluid coupling co-simulation verification platform, the simulation in the working condition of a consecutive double-lane change is carried out with tight coupling between vehicular dynamics and CFD. The simulation verifies the feasibility of the proposed observation method based on the multi-DOF pendulum model and free surface fluctuation sensor. Its advantages over the conventional open-loop calculation method include providing reliable observation for the state observation of the controllers and ensuring the safety of practical application of control algorithms under extremities.

It is important to note that this study focuses solely on liquid sloshing in the roll plane and neglects the longitudinal flow of liquid in the tank and its impact on lateral sloshing under conditions involving significant acceleration and/or braking. Future research should consider the coupling of lateral and longitudinal sloshing in modeling, prototype the fluctuation sensor, and conduct model car tests with an integrated control algorithm for rollover prevention and trajectory tracking based on model predictive control.

## Figures and Tables

**Figure 1 sensors-23-08831-f001:**
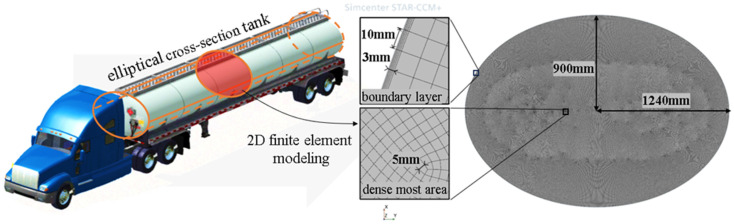
Elliptical cross-section tank and its 2D mesh modeling in Star-CCM+.

**Figure 2 sensors-23-08831-f002:**
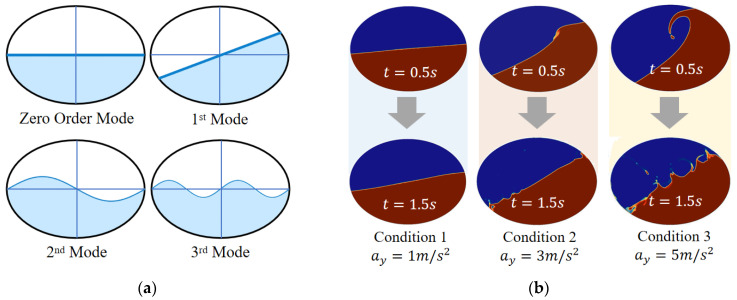
(**a**) Different sloshing mode of liquid in a tank; (**b**) CFD simulation under step excitation $a_{y}=1,3,5\;m/s^{2}$; (**c**) lateral forces from liquid to tank; and (**d**) FFT transformation of side forces.

**Figure 3 sensors-23-08831-f003:**
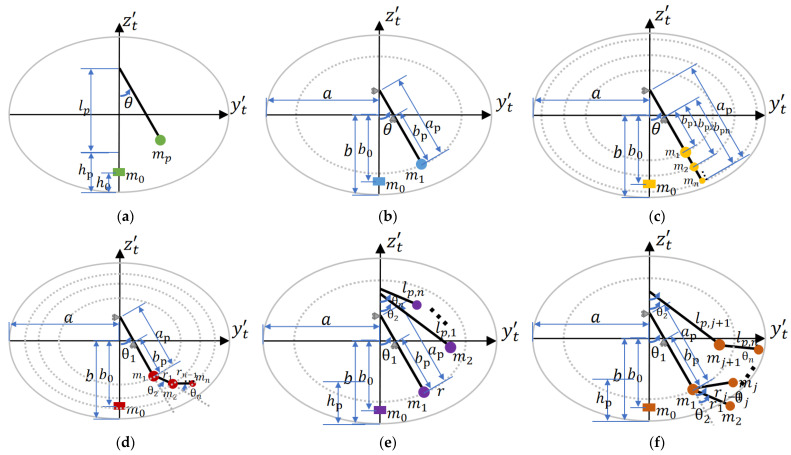
Examples of general pendulum models: (**a**) single pendulum, 1DOF, 2 concentrated masses (1 swinging, 1 fixed); (**b**) trammel pendulum, 1DOF, 2 concentrated masses (1 swinging, 1 fixed); (**c**) 1DOF multi-masses trammel pendulum [35,36], 1DOF, n + 1 concentrated-masses (n swinging, 1 fixed); (**d**) multi-DOF trammel pendulum, n DOF, n + 1 concentrated masses (n swinging, 1 fixed); (**e**) linear combination of SPs and TPs, n DOF, n + 1 concentrated-masses (n swinging, 1 fixed); and (**f**) non-linear combination of SPs and TPs, n DOF, n + 1 concentrated masses (n swinging, 1 fixed).

**Figure 4 sensors-23-08831-f004:**
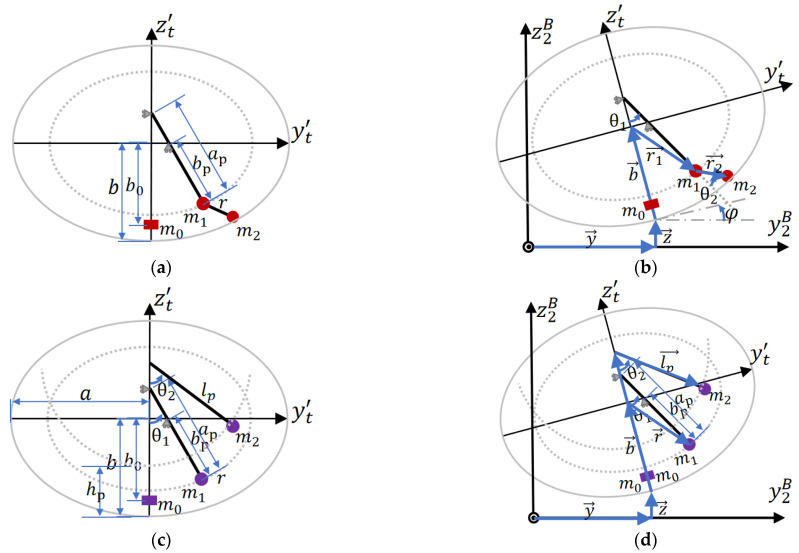
Proposed 2DOF pendulum models: DMTP and TPSP, where $o_{2}^{B}x_{2}^{B}y_{2}^{B}z_{2}^{B}$ is the coordinate fixed on the trailer body and $o_{t}^{\prime }x_{t}^{\prime }y_{t}^{\prime }z_{t}^{\prime }$ is the coordinate fixed on the tank with $o_{t}^{\prime }$ located at the center of the elliptical center. (**a**) Geometry parameters of DMTP; (**b**) degrees of freedom and dynamics of DMTP; (**c**) geometry parameters of DMTP; and (**d**) degrees of freedom and dynamics of DMTP.

**Figure 5 sensors-23-08831-f005:**
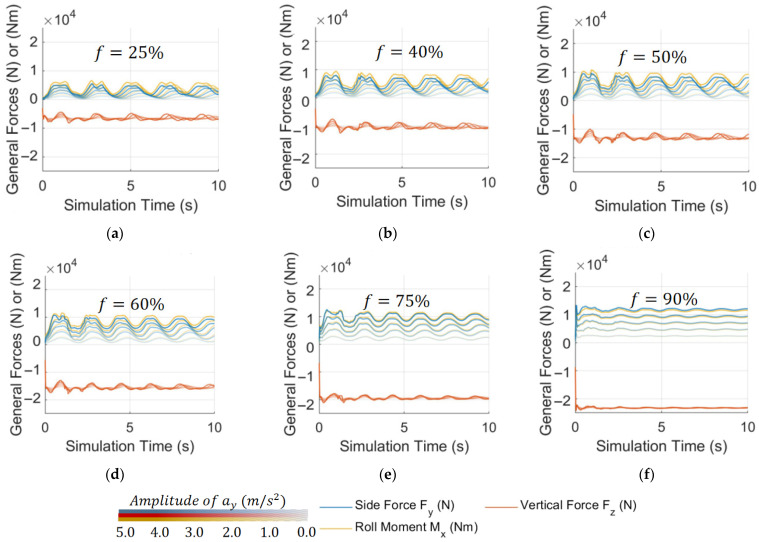
CFD simulation dataset No.1: with step excitation of lateral acceleration $a_{y}(t)=Const$, $Const=1,2,3,4,5\;m/s^{2}$ and (**a**–**f**) $f=\left[25,40,50,60,75,90\right]\%$, respectively.

**Figure 6 sensors-23-08831-f006:**
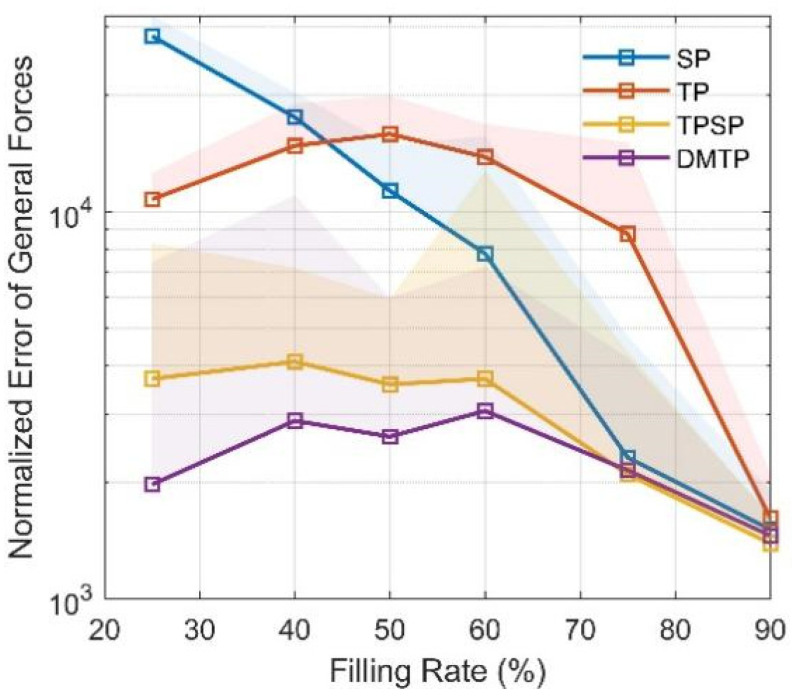
Cost function optimization results. Solid lines indicate the best result in 8 trials, and the shading indicates the distribution of the results.

**Figure 7 sensors-23-08831-f007:**
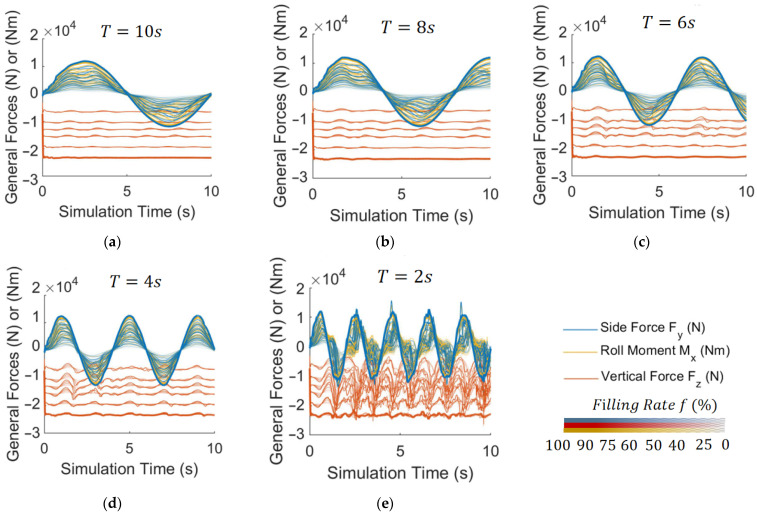
CFD simulation dataset No.2: with sinusoidal excitation of lateral acceleration, $a_{y}\left(t\right)=A\sin\left(\frac{2\pi }{T}t\right)$, $A=1,2,3,4,5\;m/s^{2}$, and (**a**–**e**) $T=\left[10,8,6,4,2\right]\;s$, respectively. $f=\left[25,40,50,60,75,90\right]\%$. A total of 150 CFD simulations.

**Figure 8 sensors-23-08831-f008:**
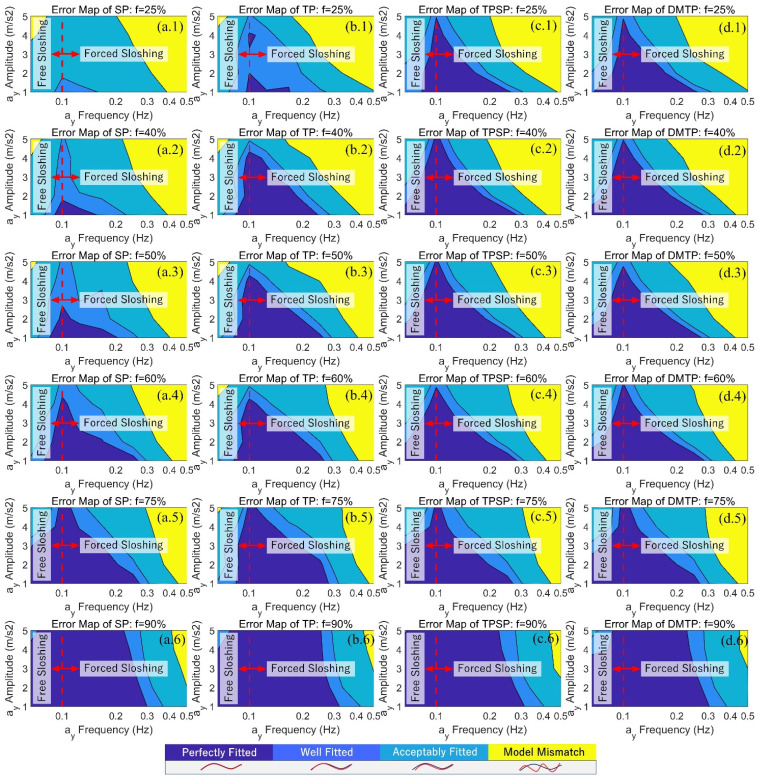
Error map of SP, TP, TPSP, and DMTP under orthogonal conditions of lateral acceleration excitation $a_{y}$: (**a.1**–**a.6**) for the SP model under different filling rates; (**b.1**–**b.6**) for the SP model; (**c.1**–**c.6**) for the TPSP model; and (**d.1**–**d.6**) for the DMTP model.

**Figure 9 sensors-23-08831-f009:**
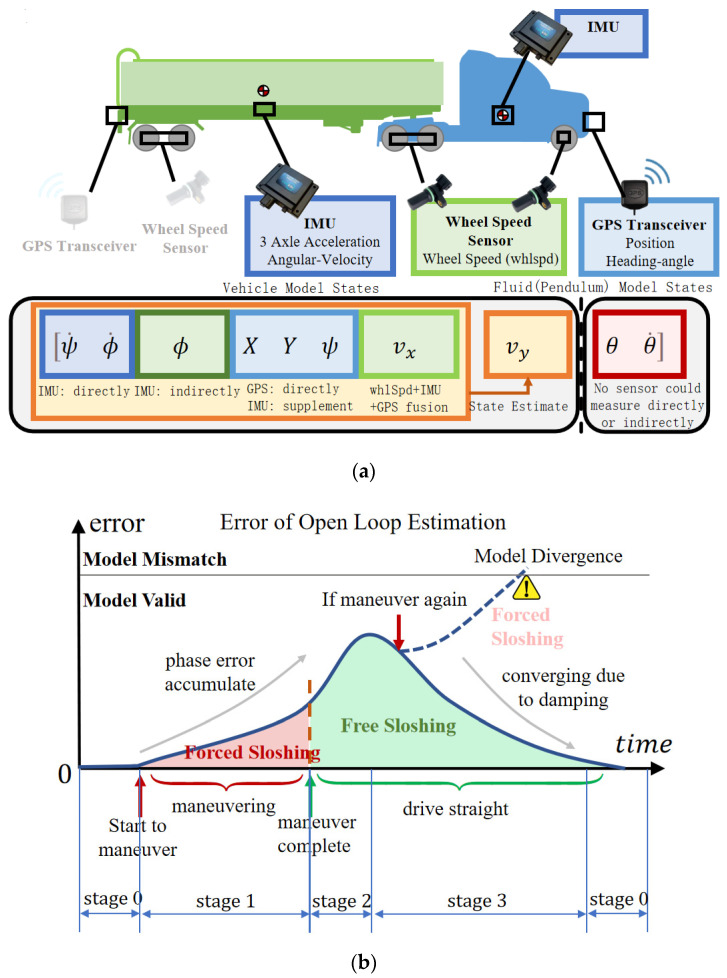
Necessity for a fluctuation sensor. (**a**) Fluid (pendulum) model-related state variables are unavailable. (**b**) Error characteristics of open loop estimation.

**Figure 10 sensors-23-08831-f010:**
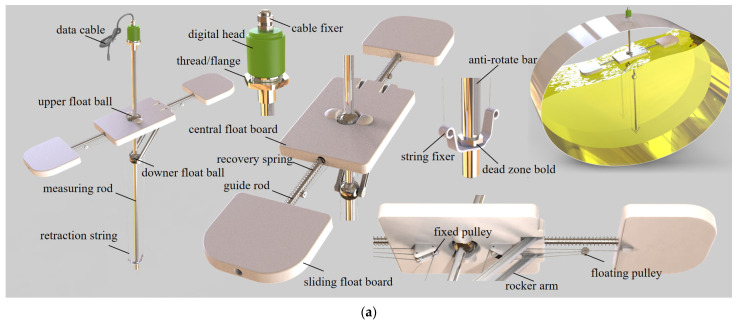
Fluctuation sensor design. (**a**) Sensor assembly and example of use. (**b**) Measurement principle. (**c**) Sliding float board size considering the measurement range. (**d**) Pulley block mechanism ensuring adaptive length adjustment.

**Figure 11 sensors-23-08831-f011:**
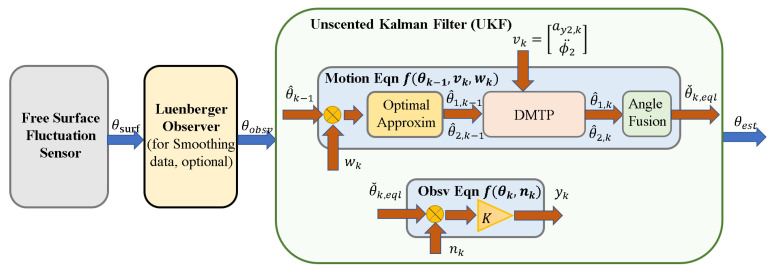
Unscented Kalman filter based on DMTP and fluctuation sensor data.

**Figure 12 sensors-23-08831-f012:**
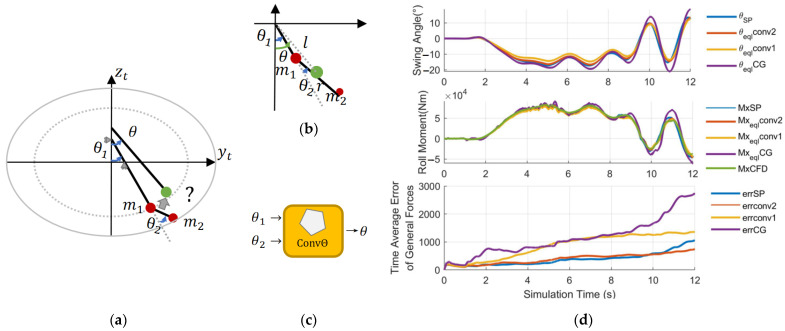
Method to convert DMTP into LSP. (**a**) Problem Description. How to convert two DOF into one: (**b**) the center of mass conversion method and (**c**) the convex combination method. (**d**) Verification in the left cornering condition. I Verification in the single-lane change condition. (**f**) Verification in the double-lane change condition. CG method in the legend correspond to (4–5), conv1 method to (7), conv2 method to (8). “?” means the conversion method is under discussion.

**Figure 13 sensors-23-08831-f013:**
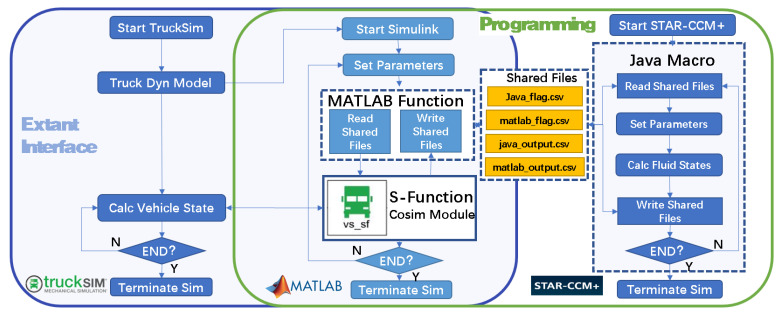
Vehicle–fluid co-simulation platform construction by bridging TruckSim and StarCCM+ by Matlab/Simulink.

**Figure 14 sensors-23-08831-f014:**
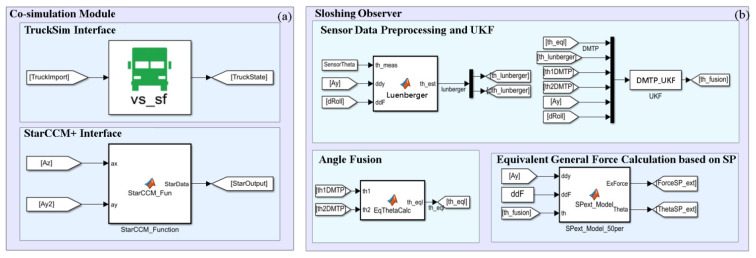
Simulation model for verification of the proposed observation method. (**a**) Vehicle–fluid-co-simulation module. (**b**) Sloshing observer.

**Figure 15 sensors-23-08831-f015:**
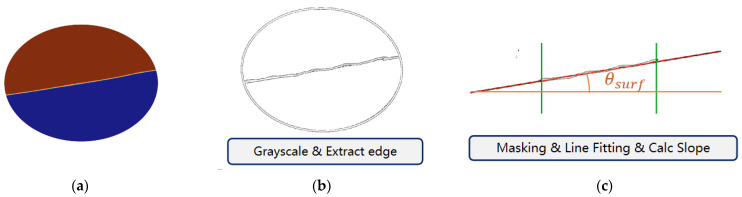
Graphical processing method to simulate a fluctuation sensor. (**a**) Original image output from Star-CCM+. (**b**) Grayscale the image and extract edges, then mask irrelevant parts. (**c**) Trim the free surface according to float board length and fit the free surface with a straight line to calculate the slope angle as $\theta_{surf}$.

**Figure 16 sensors-23-08831-f016:**
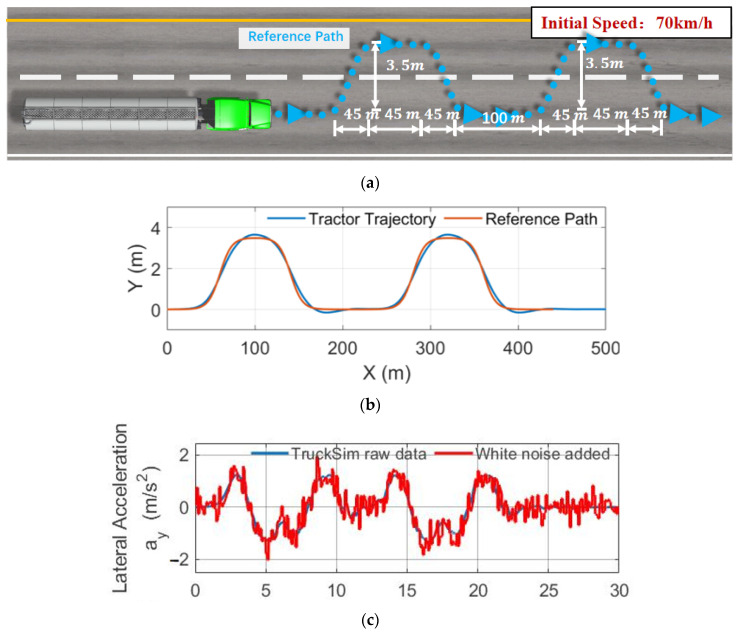
Simulation of the consecutive double-lane change condition (CDLC). (**a**) CDLC introduction. (**b**) Trajectory observations. (**c**) Observer input: lateral acceleration of the trailer. (**d**) Observer input: rolling angular velocity of the trailer. (**e**) Estimated swing angle θ; (**f**) Estimated roll moment $M_{x}$. (**g**) Time average error of $M_{x}$.

**Table 1 sensors-23-08831-t001:** Parameters for Euler phases.

Euler Phase Name	Phase	Material	Dynamic Viscosity	Density (Constant)
Gasoline	Liquid	C8H17(Gasoline)	5.0272 × 10^−4^ Pa·s	751 $kg/m^{3}$
Air	Gas	Air	1.85508 × 10^−5^ Pa·s	1.18415 $kg/m^{3}$

**Table 2 sensors-23-08831-t002:** Other continuum parameters.

Model	Settings
Wall distance	Implicit tree
Multiphase interaction	Phase interaction between phases
Euler phases	See Table 1
Mixture	Dynamic viscosity: Volume-weighted mixture
Adaptive time step	Time step provider: Free-surface CFL condition
Volume of fluid, VOF	HRIC → HRIC gradient smoothing
Dimension	Two-dimensional
Flow regime	Segregated flow
Turbulent Model	kϵ turbulence Realizable k-ϵ two-layer model Reynolds-averaged Navier–Stokes two-layer full y+ wall treatment
Gravity	Yes

**Table 3 sensors-23-08831-t003:** Spectral analysis of different sloshing models.

Slosh Mode	0 Order	Surface Incline	First	Second	Third	Higher Order	∞
Model Name							
Rigid body	√						
QS	√	√					
Linearized SP	√	√	√				
SP/TP	√	√	√	√	√		
Multi-DOF Pendulum	√	√	√	√	√	√	
FVM-CFD	√	√	√	√	√	√	
Real liquid	√	√	√	√	√	√	√

√ suggests the highest expressing ability of the model.

## Data Availability

Not applicable.

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
