# Peer review of "Observing Liquid Sloshing Based on a Multi-Degree-of-Freedom Pendulum Model and Free Surface Fluctuation Sensor"

_sensors, 2023, doi:10.3390/s23218831_

Round 1

Reviewer 1 Report

Please find attached the comments.

Author Response

Dear Reviewer:

Thank you for your precious time proofreading our manuscript, and we would like to distribute our greatest gratitude to you and your enlightening advices.

Detailed Replies are in the attachment.

Thank you again for your valuable advices!

Yours Sincerely,

Xiaojing Qi and co-authors.

Reviewer 2 Report

The manuscript, "Observing Liquid Sloshing based on Multi-DOF Pendulum Model and Free Surface Fluctuation Sensor," offers a rigorous and inventive perspective on comprehending and observing liquid sloshing dynamics in tank trucks. A commendable facet of this research is the inception of the multi-DOF pendulum models (DMTP and TPSP), which have proven to have a more adept performance in contrast to the conventional models, especially under certain filling conditions. This work was primarily simulation-based. There is no mention of physical experiments or prototype testing. However, the work includes a detailed simulation model. Nevertheless, the authors have discussed plans for prototyping the sensor and conducting model car tests.

The authors have combined advanced simulation tools in their all-encompassing and comprehensive work. Specifically, they employed MATLAB and its companion software, Simulink, to design and run sophisticated vehicle dynamics simulations. StarCCM+ was incorporated to manage and execute the computational fluid dynamics (CFD) simulations, capturing the nuanced behavior of liquid sloshing. Additionally, TruckSim was adopted to simulate vehicular motion, primarily focusing on the dynamic aspects of a semi-trailer tank truck. The integration of these platforms allowed a robust simulation experience, ensuring the consideration of both fluid and vehicular dynamics.

Integrating the free surface fluctuation sensor into the study accentuates its practical implications, especially considering its flexibility in adapting to different filling rates and its applicability in tanks containing hazardous fluids. The design choice to keep the electronic components separate from the measurement parts highlights a well-conceived strategy, addressing real-world applicative concerns.

However, one noticeable stumbling block of the paper is the extensive use of intricate mathematical expressions. While such depth is paramount for academic integrity and ensuring replicability, it considerably hampers readability. The manuscript can be intimidating for many readers, particularly those not profoundly versed in mathematical intricacies. An advisable approach would have been to place most of these detailed equations in supplementary files or appendices, allowing for a more streamlined main text. This would significantly improve the reading experience, making the core findings and innovations more accessible without sacrificing the technical depth.

Given that the research is entirely simulation-based, it carries inherent limitations. Translating these simulations into physical prototypes and subsequent real-world testing remains essential, ensuring the robustness of the introduced models and sensors beyond controlled environments.

In looking forward, the paper aptly points out the necessity to contemplate the longitudinal flow's influence on lateral sloshing and to integrate efficient control algorithms for a comprehensive solution.

While the paper presents invaluable content and a robust methodology, its dense mathematical explications detract from its overall accessibility. The authors could balance depth and readability by relegating these equations to supplementary sections. Still, its contributions to tank truck safety and control set a robust precedent for future endeavors in the field.

Therefore, in the manuscript's revised version, the authors should offer a new approach with better readability by transferring all the unnecessary mathematical expressions to a supplementary information file.

The manuscript's acceptance will depend on how the authors will reformulate the manuscript, as mentioned above.

Minor editing of English language required

Author Response

(The authors gave the same response as above.)

Round 2

Reviewer 1 Report

Please find attached the comments.

Author Response

Dear Reviewer,

We cardinally apologize for not properly citing the literature you recommended. We have made the necessary corrections and verified the accuracy of other literature citations by checking the title, authors, journal, volume number, and other relevant information with the original PDF.

On behalf of all co-authors, I would like to express my sincere gratitude to you!

Xiaojing Qi 

Reviewer 2 Report

The modifications made to the manuscript are welcome. Therefore, I accept the manuscript as is.

 Minor editing of English language required

Author Response

Dear Reviewer,

On behalf of all co-authors, I would like to express my sincere gratitude to you!

Xiaojing Qi